# Learning Graph Representations with Embedding Propagation

**Alberto García-Durán**
NEC Labs Europe
Heidelberg, Germany
`alberto.duran@neclab.eu`

**Mathias Niepert**
NEC Labs Europe
Heidelberg, Germany
`mathias.niepert@neclab.eu`

## Abstract

We propose Embedding Propagation (EP), an unsupervised learning framework for graph-structured data. EP learns vector representations of graphs by passing two types of messages between neighboring nodes. Forward messages consist of label representations such as representations of words and other attributes associated with the nodes. Backward messages consist of gradients that result from aggregating the label representations and applying a reconstruction loss. Node representations are finally computed from the representation of their labels. With significantly fewer parameters and hyperparameters an instance of EP is competitive with and often outperforms state of the art unsupervised and semi-supervised learning methods on a range of benchmark data sets.

## 1 Introduction

Graph-structured data occurs in numerous application domains such as social networks, bioinformatics, natural language processing, and relational knowledge bases. The computational problems commonly addressed in these domains are network classification [40], statistical relational learning [12, 36], link prediction [22, 24], and anomaly detection [8, 1], to name but a few. In addition, graph-based methods for unsupervised and semi-supervised learning are often applied to data sets with few labeled examples. For instance, spectral decompositions [25] and locally linear embeddings (LLE) [38] are always computed for a data set's affinity graph, that is, a graph that is first constructed using domain knowledge or some measure of similarity between data points. Novel approaches to unsupervised representation learning for graph-structured data, therefore, are important contributions and are directly applicable to a wide range of problems.

EP learns vector representations (embeddings) of graphs by passing messages between neighboring nodes. This is reminiscent of power iteration algorithms which are used for such problems as computing the PageRank for the web graph [33], running label propagation algorithms [47], performing isomorphism testing [16], and spectral clustering [25]. Whenever a computational process can be mapped to message exchanges between nodes, it is implementable in graph processing frameworks such as Pregel [29], GraphLab [23], and GraphX [44].

Graph labels represent vertex attributes such as bag of words, movie genres, categorical features, and continuous features. They are not to be confused with *class labels* of a supervised classification problem. In the EP learning framework, each vertex $v$ sends and receives two types of messages. Label representations are sent from $v$'s neighboring nodes to $v$ and are combined so as to reconstruct the representations of $v$'s labels. The gradients resulting from the application of some reconstruction loss are sent back as messages to the neighboring vertices so as to update their labels' representations and the representations of $v$'s labels. This process is repeated for a certain number of iterations or until a convergence threshold is reached. Finally, the label representations of $v$ are used to compute a representation of $v$ itself.

Despite its conceptual simplicity, we show that EP generalizes several existing machine learning methods for graph-structured data. Since EP learns embeddings by incorporating different label types (representing, for instance, text and images) it is a framework for learning with multi-modal data [31].

## 2   Previous Work

There are numerous methods for embedding learning such as multidimensional scaling (MDS) [20], Laplacian Eigenmap [3], Siamese networks [7], IsoMap [43], and LLE [38]. Most of these approaches construct an affinity graph on the data points first and then embed the graph into a low dimensional space. The corresponding optimization problems often have to be solved in closed form (for instance, due to constraints on the objective that remove degenerate solutions) which is intractable for large graphs. We discuss the relation to LLE [38] in more detail when we analyze our framework.

Graph neural networks (GNN) [39] is a general class of recursive neural networks for graphs where each node is associated with one label. Learning is performed with the Almeida-Pineda algorithm [2, 35]. The computation of the node embeddings is performed by backpropagating gradients for a supervised loss after running a recursive propagation model to convergence. In the EP framework gradients are computed and backpropagated immediately for each node. Gated graph sequence neural networks (GG-SNN) [21] modify GNN to use gated recurrent units and modern optimization techniques. Recent work on graph convolutional networks (GCNs) uses a supervised loss to inject class label information into the learned representations [18]. GCNs as well as GNNs and GG-SNNs, can be seen as instances of the Message Passing Neural Network (MPNN) framework, recently introduced in [13]. There are several significant differences between the EP and MPNN framework: (i) all instances of MPNN use a supervised loss but EP is unsupervised and, therefore, classifier agnostic; (ii) EP learns label embeddings for each of the different label types independently and combines them into a joint node representation whereas all existing instances of MPNN do not provide an explicit method for combining heterogeneous feature types. Moreover, EP's learning principle based on reconstructing each node's representation from neighboring nodes' representations is highly suitable for the inductive setting where nodes are missing during training.

Most closely related to our work is DEEPWALK [34] which applies a word embedding algorithm to random walks. The idea is that random walks (node sequences) are treated as sentences (word sequences). A SKIPGRAM [30] model is then used to learn node embeddings from the random walks. NODE2VEC [15] is identical to DEEPWALK with the exception that it explores new methods to generate random walks (the input sentences to WORD2VEC), at the cost of introducing more hyperparamenters. LINE [41] optimizes similarities between pairs of node embeddings so as to preserve their first and second-order proximity. The main advantage of EP over these approaches is its ability to incorporate graph attributes such as text and continuous features. PLANETOID [45] combines a learning objective similar to that of DEEPWALK with supervised objectives. It also incorporates bag of words associated with nodes into these supervised objectives. We show experimentally that for graph without attributes, all of the above methods learn embeddings of similar quality and that EP outperforms all other methods significantly on graphs with word labels. We can also show that EP generalizes methods that learn embeddings for multi-relational graphs such as TRANSE [5].

## 3   Embedding Propagation

A graph $G = (V, E)$ consists of a set of vertices $V$ and a set of edges $E \subseteq \{(v, w) \mid v, w \in V\}$. The approach works with directed and undirected edges as well as with multiple edge types. $\mathbf{N}(v)$ is the set of neighbors of $v$ if $G$ is undirected and the set of in-neighbors if $G$ is directed. The graph $G$ is associated with a set of $k$ label classes $\mathbf{L} = \{L_1, ..., L_k\}$ where each $L_i$ is a set of labels corresponding to label type $i$. A label is an identifier of some object and not to be confused with a class label in classification

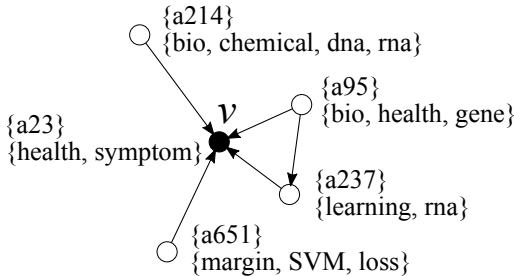

Figure 1: A fragment of a citation network.

problems. Labels allow us to represent a wide range of objects associated with the vertices such as words, movie genres, and continuous features. To illustrate the concept of label types, Figure 1

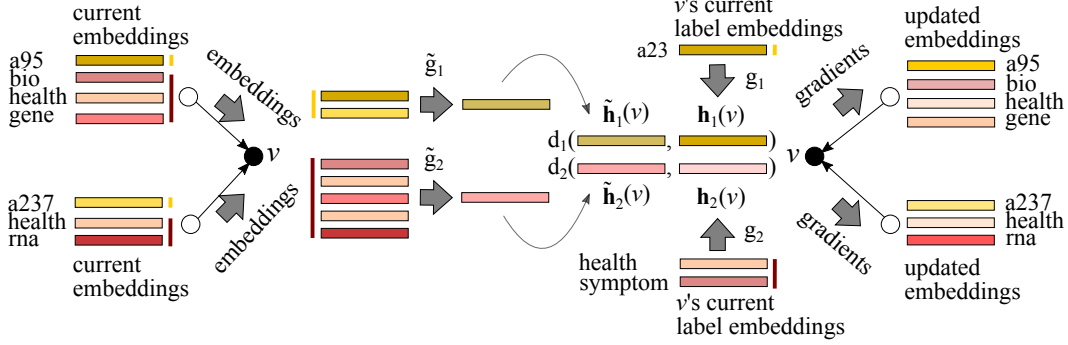

Figure 2: Illustration of the messages passed between a vertex $v$ and its neighbors for the citation network of Figure 1. First, the label embeddings are sent from the neighboring vertices to the vertex $v$ (black node). These embeddings are fed into differentiable functions $\widetilde{g}_i$. Here, there is one function for the article identifier label type (yellow shades) and one for the natural language words label type (red shades). The gradients are derived from the distances $d_i$ between (i) the output of the functions $\widetilde{g}_i$ applied to the embeddings sent from $v$'s neighbors and (ii) the output of the functions $g_i$ applied to $v$'s label embeddings. The better the output of the functions $\widetilde{g}_i$ is able to reconstruct the output of the functions $g_i$, the smaller the value of the distance measure. The gradients are the messages that are propagated back to the neighboring nodes so as to update the corresponding embedding vectors. The figure is best seen in color.

depicts a fragment of a citation network. There are two label types. One representing the unique article identifiers and the other representing the identifiers of natural language words occurring in the articles.

The functions $\mathtt{l}_i : V \to 2^{L_i}$ map every vertex in the graph to a subset of the labels $L_i$ of label type $i$. We write $\mathtt{l}(v) = \bigcup_i \mathtt{l}_i(v)$ for the set of all labels associated with vertex $v$. Moreover, we write $\mathtt{l}_i(\mathbf{N}(v)) = \{\mathtt{l}_i(u) \mid u \in \mathbf{N}(v)\}$ for the multiset of labels of type $i$ associated with the neighbors of vertex $v$.

We begin by describing the general learning framework of EP which proceeds in two steps.

- First, EP learns a vector representation for every label by passing messages along the edges of the input graph. We write $\boldsymbol{\ell}$ for the current vector representation of a label $\ell$. For labels of label type $i$, we apply a learnable embedding function $\boldsymbol{\ell} = \mathtt{f}_i(\ell)$ that maps every label $\ell$ of type $i$ to its embedding $\boldsymbol{\ell}$. The embedding functions $\mathtt{f}_i$ have to be differentiable so as to facilitate parameter updates during learning. For each label type one can chose an appropriate embedding function such as a linear embedding function for text input or a more complex convolutional network for image data.

- Second, EP computes a vector representation for each vertex $v$ from the vector representations of $v$'s labels. We write $\mathbf{v}$ for the current vector representation of a vertex $v$.

Let $v \in V$, let $i \in \{1, ..., k\}$ be a label type, and let $d_i \in \mathbb{N}$ be the size of the embedding for label type $i$. Moreover, let $\mathbf{h}_i(v) = \mathtt{g}_i\left(\{\boldsymbol{\ell} \mid \ell \in \mathtt{l}_i(v)\}\right)$ and let $\widetilde{\mathbf{h}}_i(v) = \widetilde{\mathtt{g}}_i\left(\{\boldsymbol{\ell} \mid \ell \in \mathtt{l}_i(\mathbf{N}(v))\}\right)$, where $\mathtt{g}_i$ and $\widetilde{\mathtt{g}}_i$ are differentiable functions that map multisets of $d_i$-dimensional vectors to a single $d_i$-dimensional vector. We refer to the vector $\mathbf{h}_i(v)$ as the *embedding of label type $i$* for vertex $v$ and to $\widetilde{\mathbf{h}}_i(v)$ as the *reconstruction of the embedding of label type $i$* for vertex $v$ since it is computed from the label embeddings of $v$'s neighbors. While the $\mathtt{g}_i$ and $\widetilde{\mathtt{g}}_i$ can be parameterized (typically with a neural network), in many cases they are simple parameter free functions that compute, for instance, the element-wise average or maximum of the input.

The first learning procedure is driven by the following objectives for each label type $i \in \{1, ..., k\}$

$$\min \mathcal{L}_i = \min \sum_{v \in V} \mathtt{d}_i\left(\widetilde{\mathbf{h}}_i(v), \mathbf{h}_i(v)\right), \tag{1}$$

where $\mathtt{d}_i$ is some measure of distance between $\mathbf{h}_i(v)$, the current representation of label type $i$ for vertex $v$, and its reconstruction $\widetilde{\mathbf{h}}_i(v)$. Hence, the objective of the approach is to learn the parameters

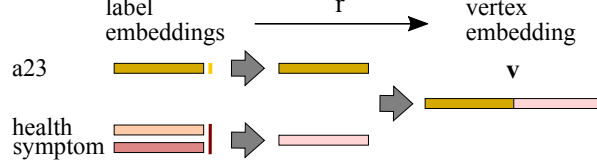

Figure 3: For each vertex $v$, the function $\mathbf{r}$ computes a vector representation of the vertex based on the vector representations of $v$'s labels.

of the functions $\mathbf{g}_i$ and $\widetilde{\mathbf{g}}_i$ (if such parameters exist) and the vector representations of the labels such that the output of $\widetilde{\mathbf{g}}_i$ applied to the type $i$ label embeddings of $v$'s neighbors is close to the output of $\mathbf{g}_i$ applied to the type $i$ label embeddings of $v$. For each vertex $v$ the messages passed to $v$ from its neighbors are the representations of their labels. The messages passed back to $v$'s neighbors are the gradients which are used to update the label embeddings. The gradients also update $v$'s label embeddings. Figure 2 illustrates the first part of the unsupervised learning framework for a part of a citation network. A representation is learned both for the article identifiers and the words occurring in the articles. The gradients are computed based on a loss between the reconstruction of the label type embeddings and their current values.

Due to the learning principle of EP, nodes that do not have any labels for label type $i$ can be assigned a new dummy label unique to the node and the label type. The representations learned for these dummy labels can then be used as part of the representation of the node itself. Hence, EP is also applicable in situations where data is missing and incomplete.

The embedding functions $\mathbf{f}_i$ can be initialized randomly or with an existing model. For instance, embedding functions for words can be initialized using word embedding algorithms [30] and those for images with pretrained CNNs [19, 11]. Initialized parameters are then refined by the application of EP. We can show empirically, however, that random initializations of the embedding functions $\mathbf{f}_i$ also lead to effective vertex embeddings.

The second step of the learning framework applies a function $\mathbf{r}$ to compute the representations of the vertex $v$ from the representations of $v$'s labels: $\mathbf{v} = \mathbf{r}\left(\{\boldsymbol{\ell} \mid \ell \in \mathbf{1}(v)\}\right)$. Here, the label embeddings and the parameters of the functions $\mathbf{g}_i$ and $\widetilde{\mathbf{g}}_i$ (if such parameters exist) remain unchanged. Figure 3 illustrates the second step of EP.

We now introduce **EP-B**, an instance of the EP framework that we have found to be highly effective for several of the typical graph-based learning problems. The instance results from setting $\mathbf{g}_i(\mathbf{H}) = \widetilde{\mathbf{g}}_i(\mathbf{H}) = \frac{1}{|\mathbf{H}|}\sum_{\mathbf{h}\in\mathbf{H}}\mathbf{h}$ for all label types $i$ and all sets of embedding vectors $\mathbf{H}$. In this case we have, for any vertex $v$ and any label type $i$,

$$\mathbf{h}_i(v) = \frac{1}{|\mathbf{1}_i(v)|}\sum_{\ell\in\mathbf{1}_i(v)}\boldsymbol{\ell}, \qquad \widetilde{\mathbf{h}}_i(v) = \frac{1}{|\mathbf{1}_i(\mathbf{N}(v))|}\sum_{u\in\mathbf{N}(v)}\sum_{\ell\in\mathbf{1}_i(u)}\boldsymbol{\ell}. \tag{2}$$

In conjunction with the above functions $\mathbf{g}_i$ and $\widetilde{\mathbf{g}}_i$, we can use the margin-based ranking loss[1]

$$\mathcal{L}_i = \sum_{v\in V}\sum_{u\in V\setminus\{v\}}\left[\gamma + \mathbf{d}_i\left(\widetilde{\mathbf{h}}_i(v), \mathbf{h}_i(v)\right) - \mathbf{d}_i\left(\widetilde{\mathbf{h}}_i(v), \mathbf{h}_i(u)\right)\right]_+, \tag{3}$$

where $\mathbf{d}_i$ is the Euclidean distance, $[x]_+$ is the positive part of $x$, and $\gamma > 0$ is a margin hyperparameter. Hence, the objective is to make the distance between $\widetilde{\mathbf{h}}_i(v)$, the reconstructed embedding of label type $i$ for vertex $v$, and $\mathbf{h}_i(v)$, the current embedding of label type $i$ for vertex $v$, smaller than the distance between $\widetilde{\mathbf{h}}_i(v)$ and $\mathbf{h}_i(u)$, the embedding of label type $i$ of a vertex $u$ different from $v$. We solve the minimization problem with gradient descent algorithms and use one node $u$ for every $v$ in each learning iteration. Despite using only first-order proximity information in the reconstruction of the label embeddings, this learning is effectively propagating embedding information across the graph: an update of a label embedding affects neighboring label embeddings which, in other updates, affects their neighboring label embeddings, and so on; hence the name of this learning framework.

Table 1: Number of parameters and hyperparameters for a graph without node attributes.

| Method | #params | #hyperparams |
|--------|---------|--------------|
| DEEPWALK [34] | $2d\|V\|$ | 4 |
| NODE2VEC [15] | $2d\|V\|$ | 6 |
| LINE [41] | $2d\|V\|$ | 2 |
| PLANETOID [45] | $\gg 2d\|V\|$ | $\geq 6$ |
| EP-B | $d\|V\|$ | 2 |

Table 2: Dataset statistics. $k$ is the number of label types.

| Dataset | $\|V\|$ | $\|E\|$ | #classes | $k$ |
|---------|---------|---------|----------|-----|
| BlogCatalog | 10,312 | 333,983 | 39 | 1 |
| PPI | 3,890 | 76,584 | 50 | 1 |
| POS | 4,777 | 184,812 | 40 | 1 |
| Cora | 2,708 | 5,429 | 7 | 2 |
| Citeseer | 3,327 | 4,732 | 6 | 2 |
| Pubmed | 19,717 | 44,338 | 3 | 2 |

Finally, a simple instance of the function $\mathtt{r}$ is a function that concatenates all the embeddings $\mathbf{h}_i(v)$ for $i \in \{1, ..., k\}$ to form one single vector representation $\mathbf{v}$ for each node $v$

$$\mathbf{v} = \mathtt{concat}\left[\mathtt{g}_1(\{\boldsymbol{\ell} \mid \ell \in \mathtt{l}_1(v)\}), ..., \mathtt{g}_k(\{\boldsymbol{\ell} \mid \ell \in \mathtt{l}_k(v)\})\right] = \mathtt{concat}\left[\mathbf{h}_1(v), ..., \mathbf{h}_k(v)\right]. \quad (4)$$

Figure 3 illustrates the working of this particular function $\mathtt{r}$. We refer to the instance of the learning framework based on the formulas (2),(3), and (4) as EP-B. The resulting vector representation of the vertices can now be used for downstream learning problems such as vertex classification, link prediction, and so on.

## 4 Formal Analysis

We now analyze the computation and model complexities of the EP framework and its connection to existing models.

### 4.1 Computational and Model Complexity

Let $G = (V, E)$ be a graph (either directed or undirected) with $k$ label types $\mathbf{L} = \{L_1, ..., L_k\}$. Moreover, let $\mathtt{lab}_{\max} = \max_{v \in V, i \in \{1,...,k\}} |\mathtt{l}_i(v)|$ be the maximum number of labels for any type and any vertex of the input graph, let $\mathtt{deg}_{\max} = \max_{v \in V} |\mathbf{N}(v)|$ be the maximum degree of the input graph, and let $\tau(n)$ be the worst-case complexity of computing any of the functions $\mathtt{g}_i$ and $\widetilde{\mathtt{g}}_i$ on $n$ input vectors of size $d_i$. Now, the worst-case complexity of one learning iteration is

$$\mathcal{O}\left(k|V|\tau(\mathtt{lab}_{\max}\mathtt{deg}_{\max})\right).$$

For an input graph without attributes, that is, where the only label type represents node identities, the worst-case complexity of one learning iteration is $\mathcal{O}(|V|\tau(\mathtt{deg}_{\max}))$. If, in addition, the complexity of the single reconstruction function is linear in the number of input vectors, the complexity is $\mathcal{O}(|V|\mathtt{deg}_{\max})$ and, hence, linear in both the number of nodes and the maximum degree of the input graph. This is the case for most aggregation functions and, in particular, for the functions $\widetilde{\mathtt{g}}_i$ and $\mathtt{g}_i$ used in EP-B, the particular instance of the learning framework defined by the formulas (2),(3), and (4). Furthermore, the average complexity is linear in the average node degree of the input graph. The worst-case complexity of EP can be limited by not exchanging messages from all neighbors but only a sampled subset of size at most $\kappa$. We explore different sampling scenarios in the experimental section.

In general, the number of parameters and hyperparameters of the learning framework depends on the parameters of the functions $\mathtt{g}_i$ and $\widetilde{\mathtt{g}}_i$, the loss functions, and the number of distinct labels of the input graph. For graphs without attributes, the only parameters of EP-B are the embedding weights and the only hyperparameters are the size of the embedding $d$ and the margin $\gamma$. Hence, the number of parameters is $d|V|$ and the number of hyperparameters is 2. Table 1 lists the parameter counts for a set of state of the art methods for learning embeddings for graphs without attributes.

### 4.2 Comparison to Existing Models

EP-B is related to locally linear embeddings (LLE) [38]. In LLE there is a single function $\widetilde{\mathtt{g}}$ which computes a linear combination of the vertex embeddings. $\widetilde{\mathtt{g}}$'s weights are learned for each vertex in a separate previous step. Hence, unlike EP-B, $\widetilde{\mathtt{g}}$ does not compute the unweighted average of the input embeddings. Moreover, LLE does not learn embeddings for the labels (attribute values) but

directly for vertices of the input graph. Finally, LLE is only feasible for graphs where each node has at most a small constant number of neighbors. LLE imposes additional constraints to avoid degenerate solutions to the objective and solves the resulting optimization problem in closed form. This is not feasible for large graphs.

In several applications, the nodes of the graphs are associated with a set of words. For instance, in citation networks, the nodes which represent individual articles can be associated with a bag of words. Every label corresponds to one of the words. Figure 1 illustrates a part of such a citation network. In this context, EP-B's learning of word embeddings is related to the CBOW model [30]. The difference is that for EP-B the context of a word is determined by the neighborhood of the vertices it is associated with and it is the embedding of the word that is reconstructed and not its one-hot encoding.

For graphs with several different edge types such as multi-relational graphs, the reconstruction functions $\widetilde{\mathbf{g}}_i$ can be made dependent on the type of the edge. For instance, one could have, for any vertex $v$ and label type $i$,

$$\widetilde{\mathbf{h}}_i(v) = \frac{1}{|\mathbf{l}_i(\mathbf{N}(v))|} \sum_{u \in \mathbf{N}(v)} \sum_{\ell \in \mathbf{l}_i(u)} \left( \boldsymbol{\ell} + \mathbf{r}_{(u,v)} \right),$$

where $\mathbf{r}_{(u,v)}$ is the vector representation corresponding to the type of the edge (the relation) from vertex $u$ to vertex $v$, and $\mathbf{h}_i(v)$ could be the average embedding of $v$'s node id labels. In combination with the margin-based ranking loss (3), this is related to embedding models for multi-relational graphs [32] such as TRANSE [5].

## 5   Experiments

The objectives of the experiments are threefold. First, we compare EP-B to the state of the art on node classification problems. Second, we visualize the learned representations. Third, we investigate the impact of an upper bound on the number of neighbors that are sending messages.

We evaluate EP with the following six commonly used benchmark data sets. BlogCatalog [46] is a graph representing the social relationships of the bloggers listed on the BlogCatalog website. The class labels represent user interests. PPI [6] is a subgraph of the protein-protein interactions for Homo Sapiens. The class labels represent biological states. POS [28] is a co-occurrence network of words appearing in the first million bytes of the Wikipedia dump. The class labels represent the Part-of-Speech (POS) tags. Cora, Citeseer and Pubmed [40] are citation networks where nodes represent documents and their corresponding bag-of-words and links represent citations. The class labels represents the main topic of the document. Whereas BlogCatalog, PPI and POS are multi-label classification problems, Cora, Citeseer and Pubmed have exactly one class label per node. Some statistics of these data sets are summarized in Table 2.

### 5.1   Set-up

The input to the node classification problem is a graph (with or without node attributes) where a fraction of the nodes is assigned a class label. The output is an assignment of class labels to the test nodes. Using the node classification data sets, we compare the performance of EP-B to the state of the art approaches DEEPWALK [34], LINE [41], NODE2VEC [15], PLANETOID [45], GCN [18], and also to the baselines WVRN [27] and MAJORITY. WVRN is a weighted relational classifier that estimates the class label of a node with a weigthed mean of its neighbors' class labels. Since all the input graphs are unweighted, WVRN assigns the class label to a node $v$ that appears most frequently in $v$'s neighborhood. MAJORITY always chooses the most frequent class labels in the training set.

For all data sets and all label types the functions $\mathbf{f}_i$ are always linear embeddings equivalent to an embedding lookup table. The dimension of the embeddings is always fixed to 128. We used this dimension for all methods which is in line with previous work such as DEEPWALK and NODE2VEC for the data sets under consideration. For EP-B, we chose the margin $\gamma$ in (3) from the set of values $[1, 5, 10, 20]$ on validation data. For all approaches except LINE, we used the hyperparameter values reported in previous work since these values were tuned to the data sets. As LINE has not been applied to the data sets before, we set its number of samples to 20 million and negative samples to 5. This means that LINE is trained on (at least) an order of magnitude more examples than all other methods.

Table 3: Multi-label classification results for BlogCatalog, POS and PPI in the transductive setting. The upper and lower part list micro and macro F1 scores, respectively.

| | BlogCatalog | | | POS | | | PPI | | |
|---|---|---|---|---|---|---|---|---|---|
| $T_r$ [%] | 10 | 50 | 90 | 10 | 50 | 90 | 10 | 50 | 90 |
| EP-B | $\gamma = 1$ | | | $\gamma = 10$ | | | $\gamma = 5$ | | |
| | $35.05 \pm 0.41$ | $\mathbf{39.44 \pm 0.29}$ | $\mathbf{40.41 \pm 1.59}$ | $\mathbf{46.97 \pm 0.36}$ | $49.52 \pm 0.48$ | $50.05 \pm 2.23$ | $\mathbf{17.82 \pm 0.77}$ | $23.30 \pm 0.37$ | $24.74 \pm 1.30$ |
| DEEPWALK | $34.48 \pm 0.40$ | $38.11 \pm 0.43$ | $38.34 \pm 1.82$ | $45.02 \pm 1.09$ | $49.10 \pm 0.52$ | $49.33 \pm 2.39$ | $17.14 \pm 0.89$ | $\mathbf{23.52 \pm 0.65}$ | $25.02 \pm 1.38$ |
| NODE2VEC | $\mathbf{35.54 \pm 0.49}$ | $39.31 \pm 0.25$ | $40.03 \pm 1.22$ | $44.66 \pm 0.92$ | $48.73 \pm 0.59$ | $49.73 \pm 2.35$ | $17.00 \pm 0.81$ | $23.31 \pm 0.62$ | $24.75 \pm 2.02$ |
| LINE | $34.83 \pm 0.39$ | $38.99 \pm 0.25$ | $38.77 \pm 1.08$ | $45.22 \pm 0.86$ | $\mathbf{51.64 \pm 0.65}$ | $\mathbf{52.28 \pm 1.87}$ | $16.55 \pm 1.50$ | $23.01 \pm 0.84$ | $\mathbf{25.28 \pm 1.68}$ |
| WVRN | $20.50 \pm 0.45$ | $30.24 \pm 0.96$ | $33.47 \pm 1.50$ | $26.07 \pm 4.35$ | $29.21 \pm 2.21$ | $33.09 \pm 2.27$ | $10.99 \pm 0.57$ | $18.14 \pm 0.60$ | $21.49 \pm 1.19$ |
| MAJORITY | $16.51 \pm 0.53$ | $16.88 \pm 0.35$ | $16.53 \pm 0.74$ | $40.40 \pm 0.62$ | $40.47 \pm 0.51$ | $40.10 \pm 2.57$ | $6.15 \pm 0.40$ | $5.94 \pm 0.66$ | $5.66 \pm 0.92$ |
| EP-B | $\gamma = 1$ | | | $\gamma = 10$ | | | $\gamma = 5$ | | |
| | $\mathbf{19.08 +- 0.78}$ | $\mathbf{25.11 \pm 0.43}$ | $\mathbf{25.97 \pm 1.25}$ | $\mathbf{8.85 \pm 0.33}$ | $10.45 \pm 0.69$ | $12.17 \pm 1.19$ | $\mathbf{13.80 \pm 0.67}$ | $\mathbf{18.96 \pm 0.43}$ | $20.36 \pm 1.42$ |
| DEEPWALK | $18.16 \pm 0.44$ | $22.65 \pm 0.49$ | $22.86 \pm 1.03$ | $8.20 \pm 0.27$ | $10.84 \pm 0.62$ | $12.23 \pm 1.38$ | $13.01 \pm 0.90$ | $18.73 \pm 0.59$ | $20.01 \pm 1.82$ |
| NODE2VEC | $\mathbf{19.08 \pm 0.52}$ | $23.97 \pm 0.58$ | $24.82 \pm 1.00$ | $8.32 \pm 0.36$ | $11.07 \pm 0.60$ | $12.11 \pm 1.93$ | $13.32 \pm 0.49$ | $18.57 \pm 0.49$ | $19.66 \pm 2.34$ |
| LINE | $18.13 \pm 0.33$ | $22.56 \pm 0.49$ | $23.00 \pm 0.92$ | $8.49 \pm 0.41$ | $\mathbf{12.43 \pm 0.81}$ | $12.40 \pm 1.18$ | $12,79 \pm 0.48$ | $18.06 \pm 0.81$ | $\mathbf{20.59 \pm 1.59}$ |
| WVRN | $10.86 \pm 0.87$ | $17.46 \pm 0.74$ | $20.10 \pm 0.98$ | $4.14 \pm 0.54$ | $4.42 \pm 0.35$ | $4.41 \pm 0.53$ | $8.60 \pm 0.57$ | $14.65 \pm 0.74$ | $17.50 \pm 1.42$ |
| MAJORITY | $2.51 \pm 0.09$ | $2.57 \pm 0.08$ | $2.53 \pm 0.31$ | $3.38 \pm 0.13$ | $3.36 \pm 0.14$ | $3.36 \pm 0.44$ | $1.58 \pm 0.25$ | $1.51 \pm 0.27$ | $1.44 \pm 0.35$ |

Table 4: Multi-label classification results for BlogCatalog, POS and PPI in the inductive setting for $T_r = 0.1$. The upper and lower part of the table list micro and macro F1 scores, respectively.

| | BlogCatalog | | POS | | PPI | |
|---|---|---|---|---|---|---|
| Removed Nodes [%] | 20 | 40 | 20 | 40 | 20 | 40 |
| EP-B | $\gamma = 10$ | $\gamma = 5$ | $\gamma = 10$ | $\gamma = 10$ | $\gamma = 10$ | $\gamma = 10$ |
| | $\mathbf{29.22 \pm 0.95}$ | $\mathbf{27.30 \pm 1.33}$ | $\mathbf{43.23 \pm 1.44}$ | $\mathbf{42.12 \pm 0.78}$ | $\mathbf{16.63 \pm 0.98}$ | $\mathbf{14.87 \pm 1.04}$ |
| DEEPWALK-I | $27.84 \pm 1.37$ | $27.14 \pm 0.99$ | $40.92 \pm 1.11$ | $41.02 \pm 0.70$ | $15.55 \pm 1.06$ | $13.99 \pm 1.18$ |
| LINE-I | $19.15 \pm 1.30$ | $19.96 \pm 2.44$ | $40.34 \pm 1.72$ | $40.08 \pm 1.64$ | $14.89 \pm 1.16$ | $13.55 \pm 0.90$ |
| WVRN | $19.36 \pm 0.59$ | $19.07 \pm 1.53$ | $23.35 \pm 0.66$ | $27.91 \pm 0.53$ | $8.83 \pm 0.91$ | $9.41 \pm 0.94$ |
| MAJORITY | $16.84 \pm 0.68$ | $16.81 \pm 0.55$ | $40.43 \pm 0.86$ | $40.59 \pm 0.55$ | $6.09 \pm 0.40$ | $6.39 \pm 0.61$ |
| EP-B | $\gamma = 10$ | $\gamma = 5$ | $\gamma = 10$ | $\gamma = 10$ | $\gamma = 10$ | $\gamma = 10$ |
| | $\mathbf{12.12 \pm 0.75}$ | $\mathbf{11.24 \pm 0.89}$ | $\mathbf{5.47 \pm 0.80}$ | $\mathbf{5.16 \pm 0.49}$ | $\mathbf{11.55 \pm 0.90}$ | $\mathbf{10.38 \pm 0.90}$ |
| DEEPWALK-I | $11.96 \pm 0.88$ | $10.91 \pm 0.95$ | $4.54 \pm 0.32$ | $4.46 \pm 0.57$ | $10.52 \pm 0.56$ | $9.69 \pm 1.14$ |
| LINE-I | $6.64 \pm 0.49$ | $6.54 \pm 1.87$ | $4.67 \pm 0.46$ | $4.24 \pm 0.52$ | $9.86 \pm 1.07$ | $9.15 \pm 0.74$ |
| WVRN | $9.45 \pm 0.65$ | $9.18 \pm 0.62$ | $3.74 \pm 0.64$ | $3.87 \pm 0.44$ | $6.90 \pm 1.02$ | $6.81 \pm 0.89$ |
| MAJORITY | $2.50 \pm 0.18$ | $2.59 \pm 0.19$ | $3.35 \pm 0.24$ | $3.27 \pm 0.15$ | $1.54 \pm 0.31$ | $1.55 \pm 0.26$ |

We did not simply copy results from previous work but used the authors' code to run all experiments again. For DEEPWALK we used the implementation provided by the authors of NODE2VEC (setting $p = 1.0$ and $q = 1.0$). We also used the other hyperparameters values for DEEPWALK reported in the NODE2VEC paper to ensure a fair comparison. We did 10 runs for each method in each of the experimental set-ups described in this section, and computed the mean and standard deviation of the corresponding evaluation metrics. We use the same sets of training, validation and test data for each method. All methods were evaluated in the transductive and inductive setting. The transductive setting is the setting where all nodes of the input graph are present during training. In the inductive setting, a certain percentage of the nodes are not part of the graph during unsupervised learning. Instead, these *removed nodes* are added after the training has concluded. The results computed for the nodes not present during unsupervised training reflect the methods ability to incorporate newly added nodes without retraining the model.

For the graphs without attributes (BlogCatalog, PPI and POS) we follow the exact same experimental procedure as in previous work [42, 34, 15]. First, the node embeddings were computed in an unsupervised fashion. Second, we sampled a fraction $T_r$ of nodes uniformly at random and used their embeddings and class labels as training data for a logistic regression classifier. The embeddings and class labels of the remaining nodes were used as test data. EP-B's margin hyperparameter $\gamma$ was chosen by 3-fold cross validation for $T_r = 0.1$ once. The resulting margin $\gamma$ was used for the same data set and for all other values of $T_r$. For each method, we use 3-fold cross validation to determine the L2 regularization parameter for the logistic regression classifier from the values $[0.01, 0.1, 0.5, 1, 5, 10]$. We did this for each value of $T_r$ and the F1 macro and F1 micro scores separately. This proved to be important since the L2 regularization had a considerable impact on the performance of the methods.

For the graphs with attributes (Cora, Citeseer, Pubmed) we follow the same experimental procedure as in previous work [45]. We sample 20 nodes uniformly at random for each class as training data, 1000 nodes as test data, and a different 1000 nodes as validation data. In the transductive setting, unsupervised training was performed on the entire graph. In the inductive setting, the 1000 test nodes were removed from the graph before training. The hyperparameter values of GCN for these same data sets in the transductive setting are reported in [18]; we used these values for both the transductive and inductive setting. For EP-B, LINE and DEEPWALK, the learned node embeddings for the 20 nodes per class label were fed to a one-vs-rest logistic regression classifier with L2 regularization. We

Table 5: Classification accuracy for Cora, Citeseer, and Pubmed. (Left) The upper and lower part of the table list the results for the transuctive and inductive setting, respectively. (Right) Results for the transductive setting where the directionality of the edges is taken into account.

| Method | Cora | Citeseer | Pubmed |
|---|---|---|---|
| EP-B | $\gamma = 20$ <br> $78.05 \pm 1.49$ | $\gamma = 10$ <br> $\mathbf{71.01 \pm 1.35}$ | $\gamma = 1$ <br> $\mathbf{79.56 \pm 2.10}$ |
| DW+BoW | $76.15 \pm 2.06$ | $61.87 \pm 2.30$ | $77.82 \pm 2.19$ |
| PLANETOID-T | $71.90 \pm 5.33$ | $58.58 \pm 6.35$ | $74.49 \pm 4.95$ |
| GCN | $\mathbf{79.59 \pm 2.02}$ | $69.21 \pm 1.25$ | $77.32 \pm 2.66$ |
| DEEPWALK | $71.11 \pm 2.70$ | $47.60 \pm 2.34$ | $73.49 \pm 3.00$ |
| BOW FEAT | $58.63 \pm 0.68$ | $58.07 \pm 1.72$ | $70.49 \pm 2.89$ |
| EP-B | $\gamma = 5$ <br> $\mathbf{73.09 \pm 1.75}$ | $\gamma = 5$ <br> $\mathbf{68.61 \pm 1.69}$ | $\gamma = 1$ <br> $\mathbf{79.94 \pm 2.30}$ |
| DW-I+BoW | $68.35 \pm 1.70$ | $59.47 \pm 2.48$ | $74.87 \pm 1.23$ |
| PLANETOID-I | $64.80 \pm 3.70$ | $61.97 \pm 3.82$ | $75.73 \pm 4.21$ |
| GCN-I | $67.76 \pm 2.11$ | $63.40 \pm 0.98$ | $73.47 \pm 2.48$ |
| BOW FEAT | $58.63 \pm 0.68$ | $58.07 \pm 1.72$ | $70.49 \pm 2.89$ |

| Method | Cora | Citeseer | Pubmed |
|---|---|---|---|
| EP-B | $\gamma = 20$ <br> $\mathbf{77.31 \pm 1.43}$ | $\gamma = 5$ <br> $\mathbf{70.21 \pm 1.17}$ | $\gamma = 1$ <br> $\mathbf{78.77 \pm 2.06}$ |
| DEEPWALK | $14.82 \pm 2.15$ | $15.79 \pm 3.58$ | $32.82 \pm 2.12$ |

chose the best value for EP-B's margins and the L2 regularizer on the validation set from the values $[0.01, 0.1, 0.5, 1, 5, 10]$. The same was done for the baselines DW+BoW and BOW FEAT. Since PLANETOID jointly optimizes an unsupervised and supervised loss, we applied the learned models directly to classify the nodes. The authors of PLANETOID did not report the number of learning iterations, so we ensured the training had converged. This was the case after 5000, 5000, and 20000 training steps for Cora, Citeseer, and Pubmed, respectively. For EP-B we used ADAM [17] to learn the parameters in a mini-batch setting with a learning rate of $0.001$. A single learning epoch iterates through all nodes of the input graph and we fixed the number of epochs to 200 and the mini-batch size to 64. In all cases, the parameteres were initilized following [14] and the learning always converged. EP was implemented with the Theano [4] wrapper Keras [9]. We used the logistic regression classifier from LibLinear [10]. All experiments were run on commodity hardware with 128GB RAM, a single 2.8 GHz CPU, and a TitanX GPU.

## 5.2 Results

The results for BlogCatalog, POS and PPI in the transductive setting are listed in Table 3. The best results are always indicated in bold. We observe that EP-B tends to have the best F1 scores, with the additional aforementioned advantage of fewer parameters and hyperparameters to tune. Even though we use the hyperparameter values reported in NODE2VEC, we do not observe significant differences to DEEPWALK. This is contrary to earlier findings [15]. We conjecture that validating the L2 regularization of the logistic regression classifier is crucial and might not have been performed in some earlier work. The F1 scores of EP-B, DEEPWALK, LINE, and NODE2VEC are significantly higher than those of the baselines WVRN and MAJORITY. The results for the same data sets in the inductive setting are listed in Table 4 for different percentages of nodes removed before unsupervised training. EP reconstructs label embeddings from the embeddings of labels of neighboring nodes.

Hence, with EP-B we can directly use the concatenation of the reconstructed embedding $\tilde{\mathbf{h}}_i(v)$ as the node embedding for each of the nodes $v$ that were not part of the graph during training. For DEEPWALK and LINE we computed the embeddings of those nodes that were removed during training by averaging the embeddings of neighboring nodes; we indicate this by the suffix I. EP-B outperforms all these methods in the inductive setting.

The results for the data sets Cora, Citeseer and Pubmed are listed in Table 5. Since these data sets have bag of words associated with nodes, we include the baseline method DW+BoW. DW+BoW concatenates the embedding of a node learned by DEEPWALK with a vector that encodes the bag of words of the node. PLANETOID-T and PLANETOID-I are the transductive and inductive formulation of PLANETOID [45]. GCN-I is an inductive variant of GCN [18] where edges from training to test nodes are removed from the graph but those from test nodes to training nodes are not. Contrary to other methods, EP-B's F1 scores on the transductive and inductive setting are very similar, demonstrating its suitability for the inductive setting. DEEPWALK cannot make use of the word labels but we included it in the evaluation to investigate to what extent the word labels improve the performance of the other methods. The baseline BOW FEAT trains a logistic regression classifier on the binary vectors encoding the bag of words of each node. EP-B significantly outperforms all existing approaches in both the transductive and inductive setting on all three data sets with one

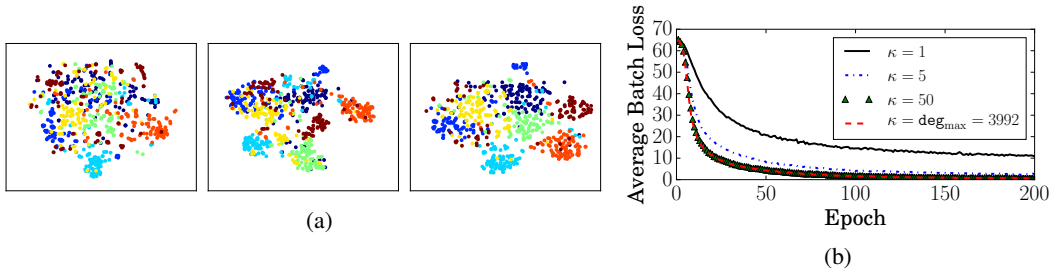

Figure 4: (a) The plot visualizes embeddings for the Cora data set learned from node identity labels only (left), word labels only (center), and from the combination of the two (right). The Silhouette score is from left to right 0.008, 0.107 and 0.158. (b) Average batch loss vs. number of epochs for different values of the parameter $\kappa$ for the BlogCatalog data set.

exception: for the transductive setting on Cora GCN achieves a higher accuracy. Both PLANETOID-T and DW+BOW do not take full advantage of the information given by the bag of words, since the encoding of the bag of words is only exposed to the respective models for nodes with class labels and, therefore, only for a small fraction of nodes in the graph. This could also explain PLANETOID-T's high standard deviation since some nodes might be associated with words that occur in the test data but which might not have been encountered during training. This would lead to misclassifications of these nodes.

Figure 4 depicts a visualization of the learned embeddings for the Cora citation network by applying t-sne [26] to the 128-dimensional embeddings generated by EP-B. Both qualitatively and quantitatively – as demonstrated by the Silhouette score [37] that measures clustering quality – it shows EP-B's ability to learn and combine embeddings of several label types.

Up until now, we did not take into account the direction of the edges, that is, we treated all graphs as undirected. Citation networks, however, are intrinsically directed. The right part of Table 5 shows the performance of EP-B and DEEPWALK when the edge directions are considered. For EP this means label representations are only sent along the directed edges. For DEEPWALK this means that the generated random walks are directed walks. While we observe a significant performance deterioration for DEEPWALK, the accuracy of EP-B does not change significantly. This demonstrates that EP is also applicable when edge directions are taken into account.

For densely connected graphs with a high average node degree, it is beneficial to limit the number of neighbors that send label representations in each learning step. This can be accomplished by sampling a subset of at most size $\kappa$ from the set of all neighbors and to send messages only from the sampled nodes. We evaluated the impact of this strategy by varying the parameter $\kappa$ in Figure 4. The loss is significantly higher for smaller values of $\kappa$. For $\kappa = 50$, however, the average loss is almost identical to the case where all neighbors send messages while reducing the training time per epoch by an order of magnitude (from 20s per epoch to less than 1s per epoch).

## 6 Conclusion and Future Work

Embedding Propagation (EP) is an unsupervised machine learning framework for graph-structured data. It learns label and node representations by exchanging messages between nodes. It supports arbitrary label types such as node identities, text, movie genres, and generalizes several existing approaches to graph representation learning. We have shown that EP-B, a simple instance of EP, is competitive with and often outperforms state of the art methods while having fewer parameters and/or hyperparameters. We believe that EP's crucial advantage over existing methods is its ability to learn label type representations and to combine these label type representations into a joint vertex embedding.

Direction of future research include the combination of EP with multitask learning, that is, learning the embeddings of labels and nodes guided by both an unsupervised loss and a supervised loss defined with respect to different tasks; a variant of EP that incorporates image and sequence data; and the integration of EP with an existing distributed graph processing framework. One might also want to investigate the application of the EP framework to multi-relational graphs.

## Footnotes

[1]Directly minimizing Equation (1) could lead to degenerate solutions.

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
