[Reviews · NeurIPS 2017]

Reviewer 1



The authors introduce embedding propagation (EP), a new message-passing method for learning representations of attributed vertices in graphs. EP computes vector representations of nodes from the 'labels' (sparse features) associated with nodes and their neighborhood. The learning of these representations is facilitated by two different types of messages sent along edges: a 'forward' message that sends the current representation of the node, and a 'backward' message that passes back the gradients of some differentiable reconstruction loss. The authors report results that are competitive with or outperform baseline representation learning methods such as deepwalk and node2vec. Quality: The quality of the paper is high. The experimental technique is clearly described, appears sound, and the authors report results that are competitive with strong baseline methods. In addition, the authors provide clear theoretical analysis of the worst-case complexity of their method. Clarity: The paper is generally clearly written and well-organized. The proposed model is clearly delineated. Originality: The work is moderately original - the authors draw inspiration from message-passing inference algorithms to improve on existing graph embedding methods. Significance: On the positive side, the proposed technique is creative and likely to be of interest to the NIPS community. Furthermore, the authors report results that are competitive or better when compared with strong baseline methods, and the computational complexity of the technique is low (linear in both the maximum degree of the graph and the number of nodes). However, while the results are competitive, it is unclear which method (if any) is offering the best performance. This is a little troubling because EP is constructing the representations using both node features and graph structure while the baselines only consider graph structure itself, and this raises the question of whether EP is adequately capturing the node features when learning the node representation. Overall, I believe that the paper is significant enough to warrant publication at NIPS. Some questions and suggestions for improvement for the authors: - It seems like this method could be implemented in a distributed asynchronous manner like standard belief propagation. If so, the scalability properties are nice - linear computational complexity and a low memory footprint due to the sparse feature input and low-dimensional dense representations. This could be a good area to explore. - It seems like there are really two representations being learned: a representation for a node's labels, and an overall representation for the node that takes the label representation of the node and the representations of the neighboring nodes into account. How necessary is it to learn a representation for the node labels as a part of the pipeline? Would it be possible to just use the raw sparse features (node labels)? If so, how would that affect performance? - Is it possible to learn this representation in a semi-supervised manner? - Learning curves would be nice. If the method is using fewer parameters, perhaps it's less inclined to overfit in a data-poor training regime? Overall impression: A good paper that proposes an interesting technique that gives solid (although not spectacular) results. An interesting idea for others to build on. Accept.

Reviewer 2



The paper presented a two-step approach for finding node embeddings of graphs, where each node may be associated with multiple labels of different types. The overall idea is to first find label embeddings based on the graph structure, and then aggregate those label embeddings to obtain node embeddings. The proposed method shows consistent improvement over several other graph embedding algorithms when multiple types of node labels are available. This is a nice addition to existing graph embedding algorithms which are either unsupervised or only consider a single label type for semi-supervised learning tasks. The method seems useful as many real-world graphs are associated with heterogeneous label types. The proposed framework also offers a unified view of several existing works, though the authors chose to use very naive aggregation strategies to implement each of the two steps to demonstrate the concept. The authors interpret their method from the label propagation perspective throughout the paper, which is corresponding to a single gradient step of some implicit objective function. I figure it would be informative to explicitly derive the underlying learning objective and compare it with other existing works. In fact, almost all graph embedding algorithms can be viewed as some sort of embedding propagation if we look at their gradient update rules. Some recent strong baselines were mentioned in the literature review but not included in the experiment, e.g. GCN [1]. 1. Kipf, Thomas N., and Max Welling. "Semi-supervised classification with graph convolutional networks." arXiv preprint arXiv:1609.02907 (2016).

Reviewer 3



The paper proposes a new framework for graph embeddings by propagating node embeddings along the observed edges in a graph. The main idea of the framework is to learn embeddings of node labels such that the embedding of all labels of a node is close to the embedding of its neighbours. The paper is generally written well and the presented concepts are mostly good to understand. The proposed model seems reasonable and technically sound. Furthermore, it shows promising results in the experimental evaluation where it is compared to standard graph embeddings on benchmark datasets. However, I am concerned with the lack of novelty in the paper (whose main contribution is the embedding propagation framework) as the idea to describe graph embeddings via a message passing/propagation framework has been proposed previously. Specifically, the recently introduced framework of Message Passing Neural Networks (MPNN) [1] seem very closely related to the framework described in Section 3. It seems that the embedding propagation framework can be described as a submodel of MPNN, where MPNN's M_t corresponds to the distance of a node embedding and its neighbor's embeddings. An interesting novel aspect of the paper is that it considers multiple labels/embeddings for nodes as this is not commonly done in standard models but has interesting applications also beyond graph embeddings. Unfortunately, the paper doesn't discuss the approach that it takes to this problem and its motivation or advantages in further detail. In summary, the paper introduces an interesting method to learn graph embeddings which shows promising experimental results. However, since the general ideas are well known and a closely related framework has recently been introduced, the novel contributions seem limited. Further comments: - I am not sure why the paper refers to h-tilda as reconstruction? The method seems to maximize the similarity of the embeddings, but doesn't attempt any explicit reconstruction of the node/neighborhood/graph. - In what sense are gradients "propagated"? It seems to me the method is doing standard gradient descent? - The term "reward" in Figure 2 is somewhat confusing as it is typically used for reinforcement-type models. - l143: This propagation property is not particularly surprising as it is common to all graph embeddings that jointly optimize embeddings of nodes such that the global link structure is captured. [1] Gilmer et al. "Neural Message Passing for Quantum Chemistry", ICML, 2017.